# A Dataset for Answering Time-Sensitive Questions

**Wenhu Chen, Xinyi Wang, William Yang Wang**
Department of Computer Science
University of California, Santa Barbara
wenhuchen@ucsb.edu, xinyi_wang@ucsb.edu, william@cs.ucsb.edu

## Abstract

Time is an important dimension in our physical world. Lots of facts can evolve with respect to time. For example, the U.S. President might change every four years. Therefore, it is important to consider the time dimension and empower the existing QA models to reason over time. However, the existing QA datasets contain rather few time-sensitive questions, hence not suitable for diagnosing or benchmarking the model's temporal reasoning capability. In order to promote research in this direction, we propose to construct a time-sensitive QA dataset. The dataset is constructed by 1) mining time-evolving facts from WikiData and aligning them to their corresponding Wikipedia page, 2) employing crowd workers to verify and calibrate these noisy facts, 3) generating question-answer pairs based on the annotated time-sensitive facts. Our dataset poses challenges in the aspect of both temporal understanding and temporal reasoning. We evaluate different SoTA long-document QA systems like BigBird and FiD on our dataset. The best-performing model FiD can only achieve 46% accuracy, still far behind the human performance of 87%. We demonstrate that these models are still lacking the ability to perform consistent temporal reasoning. Therefore, we believe that our dataset could serve as a benchmark to develop NLP models more sensitive to temporal shifts. The dataset and code are released in `https://github.com/wenhuchen/Time-Sensitive-QA`.

## 1 Introduction

As time evolves, many facts will evolve along with it, such as 'the U.S. President', 'Home Team of Lebron James', etc. Understanding the scope and interval of knowledge is an essential task studied by previous literature [Allen, 1983]. The fact evolution is commonly reflected in our daily text corpora like Wikipedia or Daily News. For example, the Wikipedia of 'Lebron James'[1] covers the whole evolution of his home team. The temporal transition of these facts is normally scattered across the long document in very diverse expressions, either represented explicitly or implicitly. Such characteristics pose great challenges to the existing NLP models. For example, a user might pose a question like 'Which team did Lebron James play for in 2007?'. The only valid evidence in Wikipedia is 'Cleveland Cavaliers (2003–2010)'. To answer this question, the model is required to perform temporal reasoning. We simulate these time-sensitive trivia questions and present them to the current state-of-art QA models [Zaheer et al., 2020, Izacard and Grave, 2020] trained on large-scale datasets. These models can only achieve a compromised 27% accuracy, much lower than their performance on Natural Questions [Kwiatkowski et al., 2019] with 64% accuracy and SQuAD [Rajpurkar et al., 2016, 2018] with 90% accuracy. This gap indicates the difficulties to handle temporal questions.

In this paper, we are specifically interested in handling these temporal questions. We formally define these questions as **time-sensitive questions** based on the following criterion: a) the question contains

---

[1] `https://en.wikipedia.org/wiki/LeBron_James`

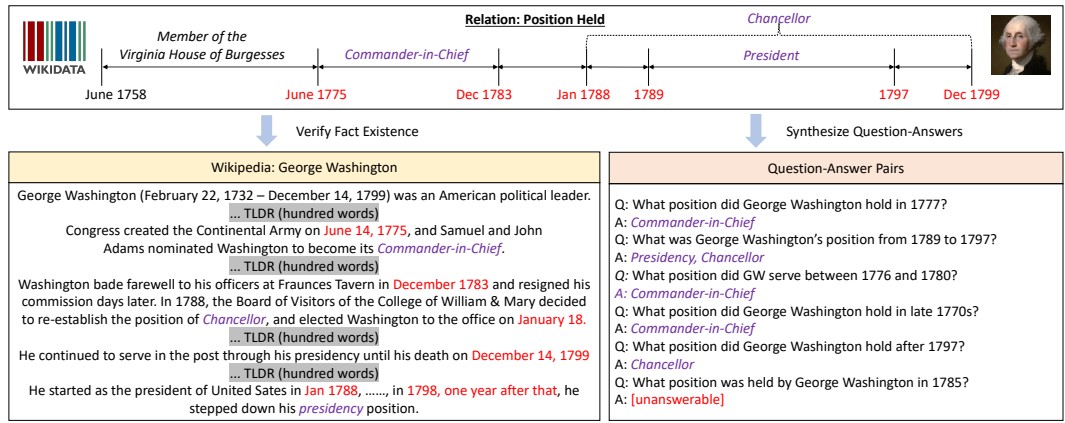

Figure 1: Time Sensitive Questions from TimeQA.

a [time specifier] like 'in 2007' or 'before 2010', b) modifying the [time specifier] will lead to answer change. c) such questions require temporal reasoning. We found that these time-sensitive questions despite their ubiquity are under-studied in the existing QA datasets. For example, our human study reveals that Natural Questions [Kwiatkowski et al., 2019] only contains less than 5% of questions with [time specifier]. In SQuAD [Rajpurkar et al., 2016, 2018], though there is a larger portion of questions with [time specifier], these questions usually copy the original time-specifying phrases from the passage without requiring any temporal reasoning, thus not meeting condition c). In TriviaQA/WebQuestions/WebComplexQuestions [Berant et al., 2013, Bao et al., 2016, Joshi et al., 2017], our human study reveals that there are more questions involving time specifiers, however, most of these specifiers in these questions are not modifiable. An example is 'What is the title of the last Harry Potter novel in 2007', where 'last Harry Potter' already implies 'year=2007', therefore, the [time specifier] is redundant and not modifiable. Thus, these questions cannot be considered time-sensitive due to condition b).

The closest to ours is Tempquestions [Jia et al., 2018a,b], which investigates the temporal questions with time specifiers. However, their questions are extracted from the above-mentioned datasets, which fails to meet the condition b). Furthermore, Tempquestions studies KG-based QA instead of Text-based QA, which differentiates from our goal of understanding temporal transition in natural text. Therefore, we propose to construct our own dataset called Time-Sensitive Question Answering (TimeQA). We first identify time-evolving facts from WikiData [Vrandečić and Krötzsch, 2014], and then employ human workers to annotate the boundaries of these facts by aligning with Wikipedia passages. We synthesize diverse question-answer pairs based on the annotated time-evolving facts using diverse templates. Finally, we create two datasets (easy and hard) with two levels of difficulty, both containing 20K question-answer pairs regarding 5.5K time-evolving fact and 70 relations. The hard version is more challenging as it requires more temporal reasoning than the easy version. The example in Figure 1 showcases some examples in our TimeQA dataset. The challenges posed by our dataset is in two folds:

- **Temporal Understanding**: understand the time scope (start and end time) of facts in the long text. However, the time information can be expressed implicitly in the text, which requires temporal commonsense to understand, for example, 'during the second world war' implies 'from 1939 to 1945', 'one year after 1934' refers to 'year 1944', etc.

- **Temporal Reasoning**: reason over the temporal information in the text conditioned on the query. More formally, the model needs to understand the temporal relationship ('within', 'between', 'before', 'after', etc.) between the time presented in the query and document.

We evaluate different state-of-the-art QA models' performance on both easy and hard versions, the performance drops from 60% to 45% on the hard version, which indicates the model suffers from its incompetence to perform temporal reasoning. When comparing with human performance of 87% on TimeQA-hard, the existing neural models are still significantly lagging behind. Therefore, we believe TimeQA could serve as a valuable benchmark in studying this problem.

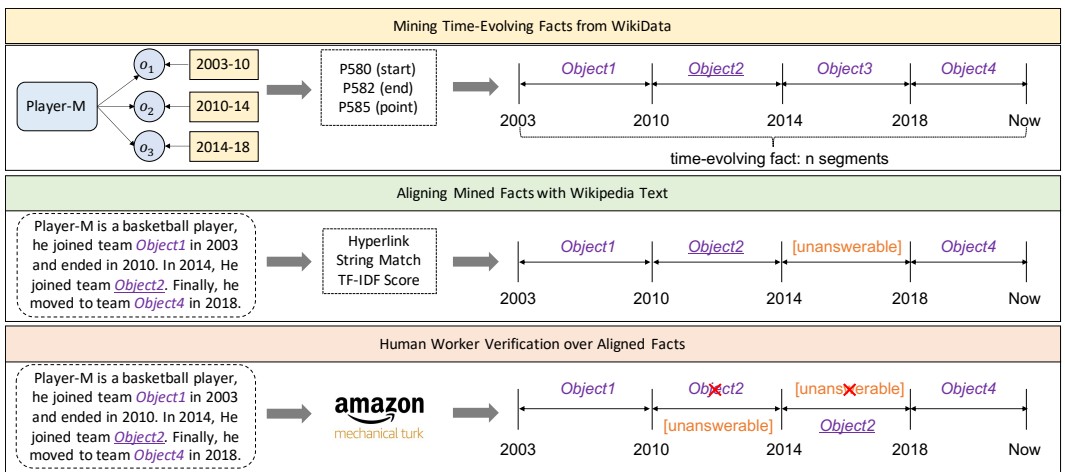

Figure 2: Fact Annotation Step: fact mining, aligning and verification.

## 2 Dataset and Problem Definition

Here we demonstrate our dataset construction pipeline, our dataset construction takes two steps: 1) fact annotation, 2) question-answer synthesizing.

### 2.1 Fact Annotation

The fact annotation takes three steps as depicted in Figure 2, which is comprised of three sub-steps, mining facts, aligning facts, and verify facts.

**Mining Time-evolving Facts from WikiData** We first identify which facts are evolving over time and through the existing annotations from WikiData [Vrandečić and Krötzsch, 2014]. Therefore, we resort to WikiData to mine these time-evolving facts, please refer to the 'Lebron James' example in https://www.wikidata.org/wiki/Q36159. As indicated in the first-row of Figure 2, we traverse the WikiData pages and select the facts with time quantifier P580 (start from), P582 (end in), and P585 (point in time) to mine interesting knowledge triples with their temporal quantifiers. We discard the triples with numeric objects like "(Denver, population of county, 13876)" because these numerical facts are unlikely to appear in Wikipedia text. We succeed to mine roughly 150K time-evolving facts in the form of $(subject, relation, \{t_1 \rightarrow t_2: object_1, t_2 \rightarrow t_3: object_2, ... t_k \rightarrow t_{k+1}: object_k\})$, where $t_k$ denotes the time boundary of $k$-th fact segment.

**Aligning Facts with Text** After mining these time-evolving facts, we need to trace them back to their Wikipedia text. We decide whether the fact segment $(subject, relation, t_k \rightarrow t_{k+1}: object_k)$ is mentioned in the Wikipedia page using the following rules: 1) $object_k$ is hyperlinked in the Wikipedia page, 2) $object_k$'s name has exact string match, 3) the TF-IDF score between the $object_k$'s name and Wikipedia text is above 40%. 4) Otherwise, the segment will be deemed "[unanswerable]". The process is depicted in the second-row of Figure 2. After this automatic tracing procedure, we discard the time-evolving facts having over than 50% "[unanswerable]" segments. On the other hand, we found that some particular relations like 'play for' are dominating the dataset. Hence, we further propose to down-sample these over-represented relations and finally identify 5.5K well-balanced facts as our candidate knowledge triples for next-step human verification.

**Human Verification** The previous step generates relatively noisy (text, time-evolving fact) pairs. There mainly exist the following sources of errors: 1) the WikiData annotation is erroneous, 2) the object is mentioned in the text, but its time boundary is not mentioned, 3) the object takes a different surface form, therefore the detected '[unanswerable]' is indeed answerable. Therefore, we propose to add human verification to clean these noisy data pairs. We provide the 5K (text,

time-evolving fact) pairs as HITs to high-quality Amazon Mechanical Turkers[2]. The workers can take the following actions: a) correcting the erroneous object, b) changing object to [unanswerable], c) changing [unanswerable] to an object in the text. The process is depicted in the third-row of Figure 2. Each HIT is paid with 1.0 dollars with an average finish time of 5 minutes. The average hourly pay is 12.0 dollars, which exceeds the income requirements proposed in human subject research protocols[3]. The annotation interface is demonstrated in the Appendix.

**Annotation Criteria** We make following guideline during annotation to help crowd-workers deal with ambivalent cases. We define extracted $t1 \rightarrow t2$ to be the correct time scope for the given fact ($subject$, $relation$, $object$) under the following conditions: a) $t_1$, $t_2$ are all explicitly mentioned in the passage, b) $t_1'$ and $t_2'$ is mentioned in the passage and $t_1' \leq t_1 < t_2 \leq t_2'$. c) $t_1'$, $t_2'$ is more coarse grained than $t_1, t_2$ (passage mentions 2018, the question mentions 2018 June), d) $t_1'$, $t_2'$ is mentioned in the passage, by combining with a reasoning function $f$, we are able to derive $t_1, t_2 = f(t_1', t_2')$. For example, 'X happens in 1987 ($t_1$), after two years, Y ...' entails 'Y happening in 1989 ($t_1$)', similarly, 'X went to university in 1987 ($t_2'$)' entails 'X graduated in 1991 ($t_2$)'. If none of the above conditions are met, we will define the time scope to be 'unknown', later on, these facts will be used to synthesize unanswerable questions.

**Quality Control** In order to harvest a high-quality dataset, we perform very detailed quality control in the collection procedure. In the interface, we will highlight the mentioned objects and time with special fonts to help the annotators identify them. We batch the HITs by their worker id to accelerate our quality assessment procedure. We sample 2 HITs from each batch and send them to our high-quality verifier to evaluate their correctness. If the sampled HITs pass our quality assessment, we will accept the other HITs within the batch. Otherwise, we will reject the whole batch. The overall acceptance rate is maintained between 85-90%.

During the verification step, roughly 41.6% of the segments are revised. The 5.5K worker-annotated examples are further filtered to obtain 5060 'golden' (text, time-evolving fact) pairs as the final release, with each fact having an average of 4 segments (time span). Out of these facts, 12% of the segments are '[unanswerable]', 7% have multiple objects, and 80% have exactly one object. The harvested facts involve over 60 different relations like 'play for', 'employee of', etc.

## 2.2 Synthesizing Question-Answer Pairs

Once we obtain the human-corrected time-evolving facts, the following step is to generate question-answer pairs from these facts.

**Main Dataset** This is the main dataset we will use throughout our paper, which relies mainly on template generation. The synthesizing procedure is described in Figure 3. For each given relation, we manually write 2-5 different templates.

We propose several common reasoning types ('in', 'between', 'before', 'after') to fill in the place-holder. For example, if we want to generate 'in (implicit)' type, we will randomly sample a time 'July 1776' within the segment '1775/6 - 1778/6' and create 'in July 1776' to fill in the placeholder.

As shown in Figure 3, we classify these types into 'easy' or 'hard' categories depending on whether the [time specifier] exactly matches the boundaries in the time-evolving axis. Since these boundaries are more likely to be mentioned in the passage explicitly, the questions with such [time specifier] on the boundary are easier to be answer based on surface form rather than temporal reasoning. In contrast, [time specifier] falling in the middle of the time span are more likely to necessitate reasoning over the implicit time information.

To better diagnose the model's capability to perform different levels of temporal reasoning, we generate two versions of TimeQA dataset. In the easy version, we only sample from 'easy' reasoning types. In the hard version, we only sample from 'hard' reasoning types. In total, we generate 20K questions (average 4 questions per time-evolving fact), for both versions. The easy questions tend to have more explicit mentions in the document, while the hard questions containing more implicit

---

[2]We select the workers from English-speaking countries, with acceptance rate over 98% and over 1000 annotation HITs being approved.

[3]https://en.wikipedia.org/wiki/Minimum_wage_in_the_United_States

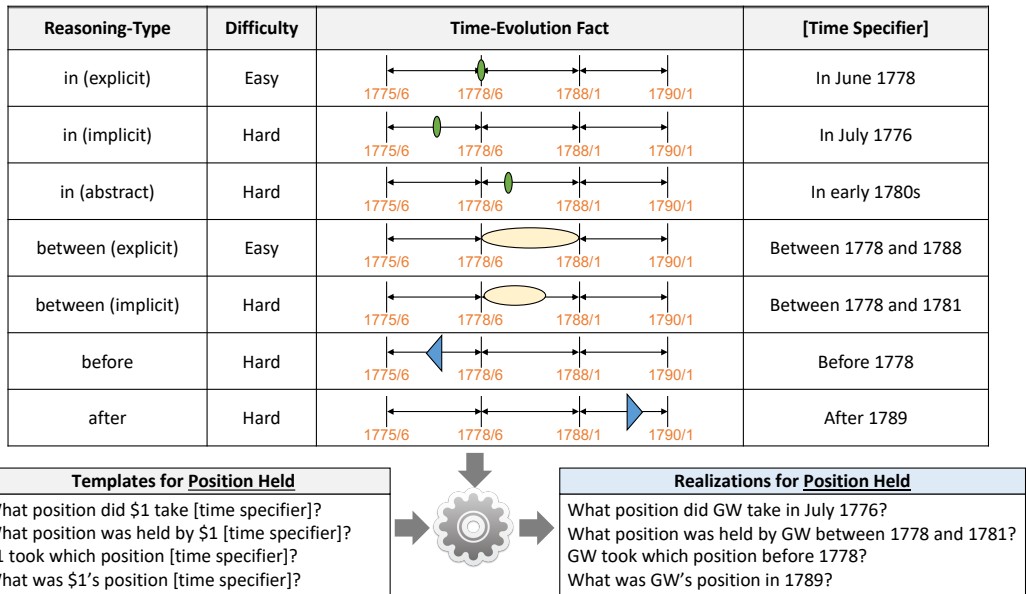

| Reasoning-Type | Difficulty | Time-Evolution Fact | [Time Specifier] |
|---|---|---|---|
| in (explicit) | Easy | | In June 1778 |
| in (implicit) | Hard | | In July 1776 |
| in (abstract) | Hard | | In early 1780s |
| between (explicit) | Easy | | Between 1778 and 1788 |
| between (implicit) | Hard | | Between 1778 and 1781 |
| before | Hard | | Before 1778 |
| after | Hard | | After 1789 |

| Templates for **Position Held** | Realizations for **Position Held** |
|---|---|
| What position did $1 take [time specifier]? | What position did GW take in July 1776? |
| What position was held by $1 [time specifier]? | What position was held by GW between 1778 and 1781? |
| $1 took which position [time specifier]? | GW took which position before 1778? |
| What was $1's position [time specifier]? | What was GW's position in 1789? |

Figure 3: QA Synthesizing Step: taking templates and annotated time-evolving fact as input to generate realistic question-answer pairs.

| Split | Mode | #Questions | #Entities | #Relations | #Answerable | #Unanswerable | #Doc-Token |
|---|---|---|---|---|---|---|---|
| Train | Easy | 14308 | 3500 | 70 | 12532 | 1776 | 1816 |
| | Hard | 14681 | 3500 | 70 | 12532 | 2149 | 1812 |
| Dev | Easy | 3021 | 748 | 52 | 2674 | 347 | 1871 |
| | Hard | 3087 | 748 | 52 | 2674 | 413 | 1871 |
| Test | Easy | 2997 | 749 | 50 | 2613 | 384 | 1864 |
| | Hard | 3078 | 749 | 50 | 2613 | 465 | 1865 |

Table 1: The dataset statistics for different splits and modes, #Doc-Token means the average number of tokens within the document.

mentions, which is more challenging for the QA models. In the following experiments, we will demonstrate the performance difference between these two versions. The comprehensive statistics of both the easy and hard versions of TimeQA are shown in Table 1. The license and privacy information for the dataset are shown in the Appendix for reference.

**Human-Paraphrased Complement** To further complement the template-generated questions, we sample a relation-balanced subset of train/test questions for human paraphrasing. In this paraphrasing process, the crowd-workers are required to rewrite the template questions to make them more natural, unambiguous, and diverse. Specifically, the human-written questions will include diverse mentions over the existing relations and entities as shown in Figure 4. We finally obtain an extra 1171 easy/hard

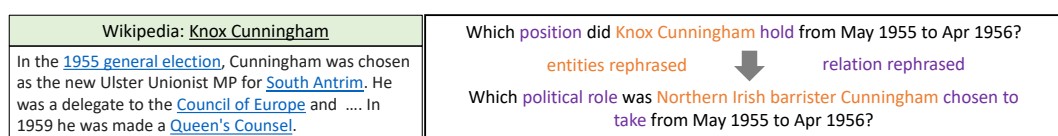

Figure 4: Crowd-Worker Paraphrasing the template questions.

questions regarding 320 time-evolving facts as our training data and 989 easy/hard questions regarding 257 time-evolving facts as our test data. The relations for this human-annotated subset are well balanced to avoid excessive over-fitting.

# 3 Models

Here we formally define the problem setup. The model is given the document $D = d_1, \cdots, d_N$ and question $Q = q_1, \cdots, q_M$, where $q_i$ and $d_i$ refers to $i$-th token in document and question with a length of N and M. The model needs to predict an answer string $\hat{A}$. To cope with the existing challenges, especially the long-term dependency, we propose to use two models BigBird [Zaheer et al., 2020] and FiD [Izacard and Grave, 2020], which are known to achieve state-of-the-art performance on the Natural Question [Kwiatkowski et al., 2019] and TriviaQA [Joshi et al., 2017]. We briefly describe their design as follows:

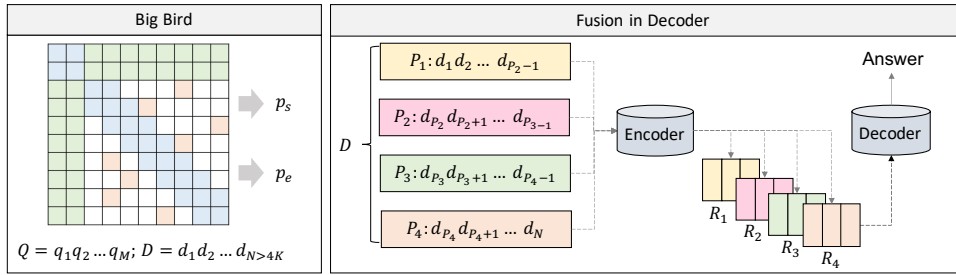

Figure 5: The extractive and generative architectures. (Left: BigBird; Right: FiD)

## 3.1 BigBird Extractive Model

This model aims to extract the start and end positions from the given sequence. The input sequence $X = (q_1, \cdots, q_M, [SEP], d_1, \cdots, d_N)$ is a concatenated sequence of question $Q$ and document $D$. Due to the length of the document, the input sequence can easily exceed 4K tokens. Therefore, BigBird [Zaheer et al., 2020] uses a more generalized attention mechanism as:

$$ATTN(X)_i = x_i + \sum_{h=1}^{H} (\sigma(Q_h(x_i)K_h(X_{N(i)})^T)) \cdot V_h(X_{N(i)}) \tag{1}$$

where $Q, K, V$ are the query, key and value functions, and $X_{N(i)}$ corresponds to the matrix formed by only stacking $x_j : j \in N(i)$ and not all the inputs. BigBird [Zaheer et al., 2020] proposes two attention mechanism: 1) a local sliding window to ensure the nodes $x_i$ attend to its local neighbors $x_{i-w/2:i+w/2}$, 2) a random subset of nodes are also being attended by $x_i$ to capture global information. The above local-global attention mechanism makes $N(i)$ a sparse matrix, which is depicted in left side of Figure 5. Such sparse attention matrix lowers the square computation cost to almost linear. After stacking multiple layers, we obtain the representation at top layer as $R_X \in \mathbb{R}^{(N+M) \times D}$ with $D$ denoting the hidden dimension. We project $R_X$ to $p_s \in \mathbb{R}^{N+M}$ and $p_e \in \mathbb{R}^{N+M}$ using the following equation, where $W_s$ and $W_e$ are both $\mathbb{R}^{D \times 1}$ learnable matrices.

$$p_s = softmax(squeeze(R_X \cdot W_s)); \quad p_e = softmax(squeeze(R_X \cdot W_e)) \tag{2}$$

During inference time, we select $i, j = argmax_{i,j}(p_s(i) \times p_e(j))$ as the start and end position of the prediction span, and the answer prediction is $\hat{A} = X_{i:j}$.

## 3.2 FiD Generative Model

This model aims to generate answer token by token in an auto-regressive fashion. In order to accommodate the long document $D$, FiD [Izacard and Grave, 2020] proposes to split it into $L$ relative short paragraphs denoted as $P_1, P_2, \cdots, P_L$, where each of the $P_i$ is shorter than a length limit of $K$ tokens (here $N \approx (K + M)L$, with $K \gg M$). The question $Q$ and $P_i$ are concatenated to build $i$-th input $\hat{P}_i$, where each of $P_i$ is fed to the T5 encoder function $f_{enc}$ to obtain their corresponding representation as $R_i \in \mathbb{R}^{(K+M) \times D}$. The encoded representations are thus concatenated to build $R = [R_1, R_2, \cdots, R_L] \in \mathbb{R}^{N \times D}$. The decoder $f_{dec}$ then attends to the concatenated $R$ to generate the answer $\hat{A}$ token by token using $p(\hat{a}_i|\hat{a}_{1:i-1}, R)$, which is described as follows:

$$R_i = f_{enc}(\hat{P}_i); \quad p(\hat{a}_i|\hat{a}_{1:i-1}, R) = softmax(f_{dec}(\hat{a}_{1:i-1}, [R_1, \cdots, R_L]) \cdot W_a) \tag{3}$$

where $W_a \in \mathbb{R}^{D \times |V|}$ is the learnable matrix to predict next-word probability over vocabulary $|V|$.

Since the decoder sequence here the attention is much shorter, the attention cost is ignorable. Thus, the total computation complexity is lowered to $O((K + M)^2 L) \approx O((K + M)N) \ll O(N^2)$. With the almost linear approximation, the generative model can handle input sequences with a length of 4K tokens. We demonstrate the model architecture on the right side of Figure 5.

# 4 Experiments

## 4.1 Experimental Setup

We conduct all the experiments based on HugginFace Transformer [Wolf et al., 2020]. The BigBird transformer checkpoints fine-tuned on NQ and TriviaQA are downloaded from `vasudevgupta/ bigbird-roberta-natural-questions` and `google/bigbird-base-trivia-itc`. These two models are based on BigBird-base with 12 layers, 12 attention heads, and a hidden dimension of 768. The maximum position embedding is 4096. Both models use a local block size of 64 and 3 random blocks for global attention. The FiD transformer checkpoints fine-tuned on NQ and TriviaQA are downloaded from `https://github.com/facebookresearch/FiD`. The FiD-base model also uses 12 layers of encoder/decoder with 12 attention heads. But its maximum position embedding is limited to 512. Thus, Both models are quite comparable in terms of parameter size.

We fine-tune all the models using AdamW [Loshchilov and Hutter, 2018] with a learning rate of 2e-5. We fine-tune all the models for 3 epochs and evaluate the performance after each epoch on the dev set to select the best-performing model. The models are trained on 4 Titan RTX GPU with 24G memory with a per-GPU-batch-size of 1. For the results in Table 2, Easy/Hard-Mode means we use the corresponding dataset for both training and evaluation.

## 4.2 Evaluation Metrics

We follow the HotpotQA [Yang et al., 2018] and NQ [Kwiatkowski et al., 2019] to use exact match and precision/recall and F1 score as the evaluation metrics. Assuming we have evaluation examples $\{Q^{(i)}, D^{(i)}, [A^{(i)}]\}$ for $i = 1, \cdots, n$, where $[A^{(i)}]$ is a list of annotated answer strings or NULL. Our prediction function outputs $\hat{A}^{(i)}$ as the answer, which is either a string or NULL. The exact match function $EM(\hat{A}^{(i)}, [A^{(i)}]) = max_j\{eq(A_j^{(i)}, \hat{A}^{(i)})\}$. For F1 score, we first split each answer $A^{(i)}$ into a set of individual words $\mathbb{A}^{(i)}$, and then compute their recall and precision. The formal definition is $F1(\hat{A}^{(i)}, [A^{(i)}]) = max_j\{F1(\mathbb{A}_j^{(i)}, \hat{\mathbb{A}}^{(i)})\}\}$. To place an upper bound on the metrics introduced above, we distribute the annotated test-set questions to crowd workers to let them provide span answers from the given passage. We compare their predictions with the annotated answers and report the approximated human results in Table 2 for comparison.

## 4.3 Main Results

We demonstrate our main results as Table 2. In the first block, we use BigBird (pre-trained with MLM) and FiD (initialized from T5 checkpoint) and only fine-tune them on our TimeQA training set without relying on external NQ/TriviaQA data. Since our dataset size is rather limited, the achieved performance is lower than 20%.

In the second block, we probe the performance of BigBird and FiD models fine-tuned on large-scale NQ/TriviaQA data. Since the TimeQA questions are linguistically simple and natural, and also coming from the Wikipedia domain, there exists very little distributional shift. However, these fine-tuned models are only achieving 33% under easy mode and 27% under the hard mode, much lower than their performance on NQ/TriviaQA (over 60%). Such a gap reflects concerning incompetence of these models to deal with temporal reasoning in text.

In the third block, we continue to fine-tune these pre-fine-tuned models on our TimeQA training set. This adaptive fine-tuning greatly enhances the models' capability to perform temporal reasoning. The performance can be significantly boosted to 60% under easy mode and 45% under hard mode. However, the best-performing model is still far behind the human performance, especially under hard mode (87%), which indicates large headroom for future studies.

| Model | Easy-Mode | | | | Hard-Mode | | | |
|---|---|---|---|---|---|---|---|---|
| | Dev | | Test | | Dev | | Test | |
| | EM | F1 | EM | F1 | EM | F1 | EM | F1 |
| BigBird (FT on TimeQA) | 16.4 | 27.5 | 16.3 | 27.1 | 11.4 | 20.6 | 11.9 | 20.3 |
| FiD (FT on TimeQA) | 15.9 | 27.1 | 15.7 | 28.0 | 10.7 | 19.1 | 10.3 | 19.7 |
| BigBird (FT on NQ) | 28.5 | 40.5 | 28.6 | 39.6 | 26.4 | 36.8 | 25.5 | 35.7 |
| BigBird (FT on TriviaQA) | **33.4** | **42.5** | **33.7** | **43.0** | **27.7** | **35.9** | **27.7** | **36.2** |
| FiD (FT on NQ) | 22.5 | 32.2 | 23.3 | 32.8 | 15.8 | 24.5 | 16.0 | 24.9 |
| FiD (FT on TriviaQA) | 23.2 | 34.0 | 23.2 | 33.0 | 13.6 | 22.0 | 13.1 | 21.4 |
| BigBird (FT on NQ + TimeQA) | 50.6 | 59.5 | 51.2 | 60.3 | 40.8 | 49.8 | 42.4 | 50.9 |
| BigBird (FT on TriviaQA + TimeQA) | 50.2 | 58.9 | 50.8 | 59.7 | 40.6 | 47.8 | 40.4 | 47.5 |
| FiD (FT on NQ + TimeQA) | **59.5** | **66.9** | **60.5** | **67.9** | **45.3** | **54.3** | **46.8** | **54.6** |
| FiD (FT on TriviaQA + TimeQA) | 56.4 | 64.9 | 57.5 | 65.1 | 44.4 | 52.5 | 46.2 | 53.7 |
| Human Worker | - | - | 89.0 | 93.3 | - | - | 87.0 | 91.1 |

Table 2: Main results for different models on our dataset, we report the EM/F1 scores for both dev/test set under easy and hard mode. All results are averaged by 3 runs.

| Model | Close-Domain QA | | | | Open-Domain QA | | | |
|---|---|---|---|---|---|---|---|---|
| | Test-Easy | | Test-Hard | | Test-Easy | | Test-Hard | |
| | EM | F1 | EM | F1 | EM | F1 | EM | F1 |
| BigBird (FT on NQ) | 26.1 | 35.3 | 21.3 | 30.1 | 8.3 | 11.1 | 6.3 | 8.7 |
| BigBird (FT on NQ + TimeQA) | 45.5 | 51.7 | 36.5 | 43.6 | 13.9 | 15.7 | 10.4 | 12.7 |
| BigBird (FT on NQ + TimeQA & Human) | 47.6 | 53.7 | 38.8 | 45.9 | 14.3 | 17.7 | 11.0 | 13.0 |

Table 3: Additional results on human-paraphrased split, we report the EM/F1 scores for both close-domain and open-domain settings. All results are averaged by 3 runs.

Throughout the experiments, we observe that the model is consistently getting much lower accuracy under hard mode. The significant 15% performance drop from easy to hard reflects the incompetence of the models to handle robust temporal reasoning. In contrast, humans are suffering only 2% drop under the hard mode, which indicates that humans are more robust in temporal reasoning.

## 4.4 Human-Paraphrased Results

We further provide experimental results on human-paraphrased questions in Table 3. We first evaluate the model fine-tuned on NQ to directly answer the human-paraphrased questions, we can observe a 2-4 points drop in EM score. Then we evaluate the model finetuned on NQ+TimeQA (only containing template questions). Surprisingly, the model without being trained on human-paraphrased questions can generalize very well to these human-paraphrases, suffering only 4-5% EM drop. After applying the 1K human-paraphrased training examples for adaptation, the gap is further decreased to only 2% EM score. The narrow performance gap between the realistic human-written questions and artificially synthesized questions indicates that our synthesized dataset is indeed an accurate proxy for estimating the model's capability to solve real-world time-sensitive questions.

Since our questions are highly decontextualized, i.e. the question is non-ambiguous, leading to a unique answer in the world. We follow DrQA [Chen et al., 2017] to perform open-domain QA, where we first use BM25 retriever to retrieve the most relevant passage from whole Wikipedia dump and then run BigBird to extract the answer. The best retrieval accuracy HITS@1 under easy and hard are 28.8% and 26.8%. The best end-task QA accuracy is rather low, achieving roughly 14% under Test-Easy and 11% under Test-Hard with lot of headroom for future work.

## 4.5 Model Analysis

Here we further analyze the model performance from different angles. Specifically, we demonstrate the model's score concerning different relations, and different document length as follows:

**Impact of Long-term Dependency** Here we are interested in understanding the models' capability to handle the long-term dependency in temporal reasoning. We plot the model's accuracy under different document lengths in Figure 6. As can be seen, the BigBird's performance degrades rapidly

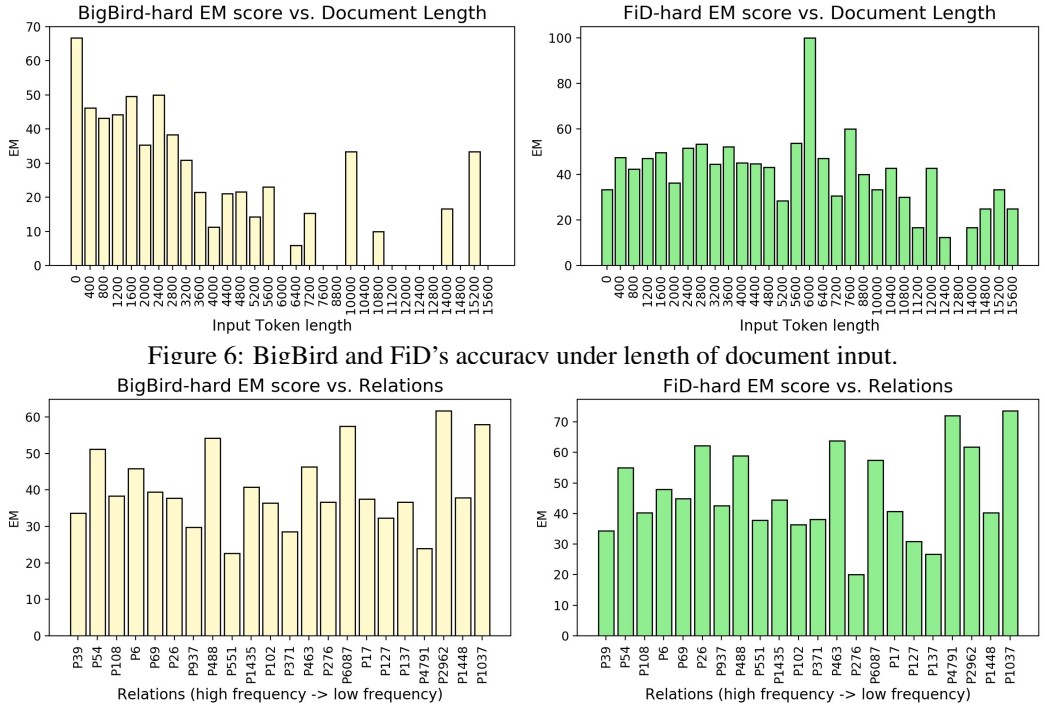

Figure 6: BigBird and FiD's accuracy under length of document input.

Figure 7: BigBird and FiD's accuracy under different relations.

as the length increases to over 5000 tokens, while the FiD's performance is quite uniformly distributed across different document lengths. This figure demonstrates that the FiD's better performance is partially attributed to its strong capability to deal with long-term dependency in temporal reasoning.

**Impact of Relation Type**    Here we are also interested in understanding the model's performance over different relations and demonstrate our findings in Figure 7. We found that the model's performance is orthogonal to the frequency of relations. For example, the relation 'P1037 (director of)' only has 30 instances, however, its performance is much higher than the relation 'P39 (position held)', which has over 500 training instances. It's mainly due to the fact that the time boundary of relation 'P1037' is more likely to be explicitly mentioned than 'P39 (position held)'.

**Consistency Analysis**    We are interested in whether the best-performing models can make consistent predictions under random perturbation of [time specifier]. For example, if a model perceives that a fact $(subject, relation, object)$ persists between $t_s$ and $t_e$, then the model should make consistent prediction for any question with [time specifier] falling within the range of $t_s$ and $t_e$. Therefore, we select the correctly predicted examples and randomly perturb the [time specifier] for 3 times. We observe how many percentages of the model predictions will remain constant/true under these random perturbations. We observe only 66% of model predictions are agnostic to these perturbations. This finding suggests that the existing QA models are not quite consistent with respect to their predictions.

### 4.6    Error Analysis

To better investigate the errors made by the QA model, we perform a detailed analysis based on the predictions from the best-performing FiD model. Under the easy-dev set, we categorize questions based on whether the [time specifier] is explicitly mentioned in the given passage. We found that two-thirds of easy-questions have [time specifier] explicitly mentioned with the average performance over 64%, while the rest easy-questions only achieve 49%. Such comparison indicates that implicit time information is a major challenge to our QA model. We further categorize the errors mainly into sources: 1) temporal reasoning: for example, '... in May 2010, one year after that' refers to 'May 2011' based on numeric addition, 2) commonsense reasoning: for example, '... in 2012 London Olympics, in the next Olympic game' refers to '2016 Olympic' based on our commonsense. 3) termination reasoning: the termination time of a fact is commonly unmentioned in a text corpus, it needs to be inferred based on the start time of the next event. For example, 'XXX joined A team in

2017, ... in 2019, B team signed a contract with XXX', we know that B team and A team are mutually exclusive, there for the termination time of A team is in 2019.

These three cases are prevalent in our daily text, which poses great challenges for the existing models. To further boost the performance on our dataset, it is vital to consider better algorithms to inject the temporal commonsense knowledge into the models.

## 5 Related Work

**Question Answering** There have been numerous efforts to tackle the machine reading comprehension problem. Different datasets like DrQA [Chen et al., 2017], TriviaQA Joshi et al. [2017], SearchQA [Dunn et al., 2017] and DROP [Dua et al., 2019] have been proposed. As the SQuAD [Rajpurkar et al., 2016] questions are relatively simple because they usually require no more than one sentence in the paragraph to answer. The following datasets further challenge the QA model's capability to handle different scenarios like open-domain, long context, multi-hop, discrete operations, etc. However, these QA datasets lack the existence the time-sensitive questions. Thus, we hope our effort could serve as a complement to the existing QA research.

**Temporal Reasoning over Knowledge Base** Understanding time evolution is an important research topic. As our world is constantly changing, it's vital to understand the time scope of world knowledge. There have been long-standing efforts to inject temporal quantifiers into the knowledge base (KB) using temporal knowledge extraction techniques [Talukdar et al., 2012b, Chang and Manning, 2012, Talukdar et al., 2012a, Wijaya et al., 2014]. Adding the temporal information into KB can empower down-stream applications like KBQA [Ahn et al., 2006, Sanampudi and Guda, 2013, Jia et al., 2018a,b, Saxena et al., 2021] to handle time-sensitive queries. Such two-step approach might suffer from cascaded errors and lead to compromised accuracy. In contrast, TimeQA aims to directly answer time-sensitive queries based on unstructured text. The new problem is more realistic yet challenging due to the high variance of human expressions over time information.

**Temporal Reasoning over Text** Recently, a contemporary dataset SituatedQA [Zhang and Choi, 2021] was also released targeting at answering open-domain time-sensitive QA. There are a few major differences: 1) SituatedQA contains more realistic queries selected from NQ dataset [Kwiatkowski et al., 2019] while TimeQA contains mostly synthesized queries (except human-paraphrased subset). 2) TimeQA contains 20K queries, while SitutatedQA only contains 4K temporal queries. 3) The hard version of TimeQA requires reasoning over implicit temporal mentions in passage, which is not emphasized in SituatedQA. Another related dataset was introduced by Dhingra et al. [2021] to diagnose whether the existing LMs are able to sensitive to changes in temporal knowledge. Their questions are mostly cloze-based probing questions under closed-book setting, while both TimeQA and SituatedQA are evaluated under open-book setting.

**Temporal Reasoning over Events** Another popular domain for temporal reasoning is event-centric tasks, which aims at understanding the textual description of real-world events [Chen et al., 2021]. There has been studies on extracting time boundaries for events [Ning et al., 2018, Wen et al., 2021, Zhang et al., 2021], reading comprehension over events [Ning et al., 2020]. These event-centric NLP tasks focus on understanding the temporal relationship (e.g. 'before', 'after', 'include', etc) between multiple events (e.g. 'snow melting', 'land slide', etc), which have inherent logical relation like 'causation', 'negation', 'inclusion', etc. Compared to TORQUE [Ning et al., 2020], there are two major differences: 1) TimeQA requires numerical reasoning over time information while TORQUE does not require it, 2) TimeQA requires modeling long-term temporal dependency in the text, while TORQUE's passages are mostly short text.

## 6 Conclusion

Though time-sensitive facts are pervasive in our daily text corpus, there has been little prior work exploring this direction. In this paper, we build the first dataset to investigate whether existing models can understand time-sensitive facts. Our experiments show that the SoTA models are still lagged behind humans in temporal reasoning. In order to empower the future NLP models to understand temporal information, different temporal-aware models need to be proposed. Finally, this paper opens up new research directions for better modeling temporal information in text representations.

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
