# OpenReview forum: "A Dataset for Answering Time-Sensitive Questions"
_NeurIPS.cc/2021/Track/Datasets_and_Benchmarks/Round2 — NeurIPS 2021 Datasets and Benchmarks Track (Round 2)_

### Official Review · Reviewer_r53P · 2021-09-18
**Good dataset, but with a bit limited diversity and coverage**

**Rating:** 7
**Confidence:** 4

**Strengths:**

* The paper sets forth a criterion for what truly challenging temporal questions should look like (Section 1 Paragraph 2). Namely, modifying the time constraint should change the answer, and requiring temporal reasoning. The authors then carried out a data collection process that fulfills these requirements.

* The dataset is collected with strict verification and quality control.


**Weaknesses:**

I’m not sure about the ontology for time-evolving facts proposed in this paper. According to Section 2.1 Paragraph 2 Line 9, the format
	 (subject, relation, {t1→t2:object1,t2→t3:object2, ... tk→tk+1:objectk})
seems to attach time modifiers exclusively to the object, but time should really be a qualifier for the entire subject-relation-object triplet.

Following the aforementioned asymmetry of temporal ontology, it seems that the questions are synthesized to ask exclusively about the object, given a subject and a time specifier. However, some time-evolving facts may be centered around the object -- for example, consider the following triples:
	({t1→t2:Tang Dynasty,t2→t3:Song Dynasty, ... tk→tk+1:Qing Dynasty}, rules, China)
And we should be able to ask questions about the subject, like “Which dynasty ruled China from t_i to t_(i+1)?” And likewise, we also might want to ask about the time specifier, such as “The Tang Dynasty ruled China between which years?” It seems that the question synthesis schema proposed in this paper lacks the ability to generate these types of questions, thereby limiting the diversity and coverage of the TimeQA dataset.

On another note, there lacks a discussion on how the granularity of time is handled, which is a crucial topic when working with temporal reasoning. For truly advanced temporal reasoning, coarse-grained and fine-grained timestamps can be mixed in the document and the question, and the dataset should reflect that. An incomplete list of question types I can think of:
* The time in the question can be more coarse-grained than in the document/fact. For example, the document may tell you “Alice played for TeamA from April 2020 to August 2020”, and the question asks “Which team did Alice play for in 2020?”
* Vice versa, The time in the question can be more fine-grained than in the document/fact. For example, the document may tell you “Alice played for TeamA in 2020”, and the question asks “Which team did Alice play for between April 2020 and August 2020?” In this case, it might be insufficient to only extract an answer span from the document, as the document’s fact has more uncertainty; the model will probably need to re-interpret the query so that it can provide grounded answers, e.g. “I know that Alice played for TeamA for at least some part of 2020.”
The WikiData likely annotates time mostly with the granularity of year, month, or day, but it’s worth to include time with other granularities (e.g. century, hour, minute) as part of the dataset, so it has more comprehensive coverage.

It is fine if handling granularity is beyond the scope of this project. But still it is worth noting your assumptions about what granularity you choose to use, and mentioning this as a limitation/future work.


**Additional Feedback:**

Your submission pdf may have been rendered in camera-ready mode and does not contain line numbers, so it’s hard to locate exact lines when communicating detailed feedback.

**Clarity:**

The paper is mostly clear and easy to follow.

The criteria for classifying proposed questions as “easy” and “hard” is not clear to me. (Section 2.2 Paragraph 2) Does the difficulty level depend on the particular instantiation of the time specifier? For example, for the same question in Figure 3, “in June 1778” would be “easy” but “in July 1776” would be “hard”?

**Correctness:**

In Section 2.1, why did the number of (text, time-evolving fact) pairs grow after human verification? You say you provide 5K such pairs to AMT workers but finally yield 5060 pairs.

In Section 3.2 Paragraph 1, there seems to be a discrepancy in the description of FiD model. Equation (3) says T5 encodes $\hat{P_i}$ into $R_i$. If so, then Line 5 should probably read “$\hat{P_i}$ is fed to the T5 encoder …” and “$R_i \in R^{(K+M) \times D}$” because the length of $Q$ and $P_i$ concatenated is $(K+M)$. Or is it the other way around -- T5 actually encodes $P_i$ only -- but then how would the question $Q$ be taken into account?


**Documentation:**

The code and data are accessible via the link to GitHub repo. Code is well documented. There are clear instructions to reproduce the benchmarks. A datasheet for datasets is filled out.

**Ethics:**

I don’t see any uncovered ethical concerns.

**Relation To Prior Work:**

Related works are thoroughly discussed. Existing work on MRC/QA does not pose real temporal reasoning challenges; whereas most of the temporal reasoning and enhancement techniques are developed around KG instead of unstructured texts. This work fills in the gap by testing deep temporal reasoning in the unstructured text setting.

**Summary And Contributions:**

This paper presents TimeQA, a QA dataset that requires models to do temporal understanding and temporal reasoning on free-form texts. It collects time-evolving facts from WikiData, aligns them with Wikipedia documents using both automatic method and human efforts, and synthesizes high-quality temporal questions. The paper also sets up benchmarks on this dataset with SOTA reading comprehension models, and finds that these models face a significant challenge in TimeQA and thus warrants future endeavors.

---

> ### Author Response · Authors · 2021-09-27
> **Author Repsonse**
>
> - Time ontology:
>
>   1. Asymmetry: This asymmetry is already partially resolved by WikiData annotation, for each relation, if the inverse relation is also 1-1 mapping, that relation will also exist in WikiData. For example, in our dataset, we have both P6087 (“coach of sports team”) and P286 (“head coach”) these two are inverse of each other. There are also other examples like ‘husband of’ and ‘wife of’, etc. In your example ({t1→t2:Tang Dynasty,t2→t3:Song Dynasty, ... tk→tk+1:Qing Dynasty}, rules, China) might have mirror annotations like (China, ruled by, {t1→t2:Tang Dynasty,t2→t3:Song Dynasty, ... tk→tk+1:Qing Dynasty}).
>
>   2. Question with time as the answer: Conceptually, it’s not hard to synthesize time-as-answer questions since we already have a data structure like (China, ruled by, {t1→t2:Tang Dynasty,t2→t3:Song Dynasty, ... tk→tk+1:Qing Dynasty}). One example will be “When did China start to be ruled by Tang?”. However, it’s likely that the answer “t1” is not explicitly mentioned in the text, the model will need to generate it from scratch. Since our motivation was to build an extractive QA dataset like NQ/SQuAD, we didn’t pursue this direction in our initial version. However, we are actively adding a subset of time-as-answer questions that has the time explicitly mentioned in the text. Since the discussion period is limited, we can’t finish all the experiments before the deadline, but we will update our paper and Github repo once these experiments are ready.
>
>   3. Annotation Time granularity: Our dataset has a mix of year and month granularity, which conforms to the original WikiData annotation. Granularity mismatch between passage and KB can sometimes happen, we specifically instruct the annotator to mind such difference. 1) if the question is finer-grained than the passage (e.g. passage states “... holds the position in 2012”, while the question suggests “What position … in Aug 2012”), the question is still answerable with the original WikiData answer or crowd worker-corrected answer. 2) if the question is more coarse-grained than passage (e.g. passage states “... holds the position in Aug 2012”, while the question suggests “What position … in 2012”), the question will be deemed answerable.  We add a section under 2.1 to discuss our criteria in our new version.
>
>   4. Diverse Time granularity: more coarse-grained annotations should be reasonable, there are indeed facts that have start/end span over centuries or decades. More fine-grained (hour/minute/second) annotations are an important direction, however, it’s quite challenging to do so with regard to Wikipedia facts, which mostly last longer than days.
>
> - Clarity
>
>   1. the difference between “easy” and “hard” is whether the [time specifier] falls on the boundary of a fact. The `easy’ question falls on the boundary, thus they are likely to be mentioned explicitly in the text. Answering these questions won’t require too much temporal reasoning, simply relying on the surface form would be enough.
>   2. For example, if there is a passage claiming “XX started working for XX in June 1778 … quit the job in Jan ​​1789”, then answering “where did XX work in June 1778” is easy, a SQuAD trained model can already handle it. However, answering something like “where did XX work in April 1788“ is more difficult since the model cannot rely on the surface form.
>   3. We have added a paragraph to section 2.2 in our new version to better explain this.
>
> - Correctness
>
>   1. The 5K is a rough number, it’s actually 5543, these facts are annotated by humans and then filtered (we manually identified some low- quality cases and use regex to detect these cases) to yield the final 5060 version. We corrected it in the revision.
>
>   2. The correct version should be R^(K+M) * D, with K much greater M. The paragraph and question are concatenated to make sure the encoding is fully aware of the questions. We corrected it in the revision. Thanks for the reminder.
>
> - Additional Feedback:
>   We have already corrected the format in our new version

---

> > ### Comment · Reviewer_r53P · 2021-09-29
> > **Thanks for your response**
> >
> > Thanks for this detailed response and the new version of paper! The concerns I raised are mostly addressed. I've updated the rating.

---

### Official Review · Reviewer_bfhP · 2021-09-19
**Interesting idea and great execution**

**Rating:** 8
**Confidence:** 4

**Strengths:**

The paper is an excellent example of how to describe the dataset construction process, starting from the analysis of the current models to the problem definition and dataset synthesis.
The analysis of results concerning the input length, the type of relation, and output consistency to the input perturbations may help future researchers deeper understand the topic's nuances.


**Weaknesses:**

I can only see one minor issue with the dataset:
1. Authors perform costly manual labeling and quality control, but the underlying text is still sourced from Wikipedia. As there are multiple wiki-based datasets, one can ask how much harder it would be to label the real-world data coming from news articles? As of now, I think this is the main limitation of the work. However, considering that only Wikipedia has a tightly coupled wikidata database, I could also see that this may be the only reasonable approach. Can you motivate the choice of Wikipedia and state that, e.g., sourcing and labeling real-world articles related to temporal reasoning is an order of magnitude more expensive, or that aligning facts with the texts helped to drastically increase the size of the dataset? These questions are not meant to downsize the quality of your work but rather to help establish the position that domain diversity is an essential factor in measuring NLP progress. I believe future readers would appreciate adding a short note to the paper to motivate why Wikipedia was chosen.


**Additional Feedback:**

See weaknesses.

**Clarity:**

The content of the paper is written clearly and in a concise manner. In addition, the structure of the article is excellent, and the graphics are informative.

**Correctness:**

The steps presented in the paper that relate to the construction of the dataset are sound and comprehensive.

**Documentation:**

The composition of the dataset, statistics are presented in the main paper. The GitHub repository is well structured and gives a helpful introduction on how to reproduce the results. The license is a permissive BSD 3-Clause "New" or "Revised" License.


**Ethics:**

It is hard to find any potential ethical concern related to the dataset itself. The hourly pay for manual workers is mentioned and is at an acceptable standard.

**Relation To Prior Work:**

Yes, the relation to prior works is a very extended and informative section.

**Summary And Contributions:**

The authors describe the lacking aspect of current SOTA NLP models, specifically, the inability to handle questions related to timing and time relations. The paper proposes a two (easy, difficult) Wikipedia-based QA dataset with variable degrees of question complexity to measure this challenging type of reasoning. The authors take extra care to curate the datasets by providing manual labels, performing quality control, down-sampling of the most dominant types of questions, providing human performance, etc. Finally, two strong models are validated as a baseline on these tasks (BigBird and FiD) to prove enough space for future improvement.
Even though the paper's main contribution is focused on temporal reasoning, the dataset may pose interesting challenges due to the fact that the input documents are long.

---

> ### Author Response · Authors · 2021-09-27
> **Author Response**
>
> First of all, thanks a lot for your recognition of our paper.
>
> - How much harder it is to annotate evolving facts from news articles.
>
>   1. Annotating news articles is definitely an interesting topic. There are a few differences that I can envision.  Since the news article dataset like CUSTOM-NEWS doesn’t have a well-established knowledge graph associated with them. It would be hard to identify what are actually the changing facts. In order to identify these changing facts, one could potentially use programmatic ways, like mining similar sentences from articles published at different times. Like there is a sentence “Trump won the election” in the “2016 Nov FOX NEWS”, and there is another sentence like “Biden won the election” in the “2020 Nov FOX NEWS”, we can kind of identify some of these cases. However, the recall and precision might be quite low, which again needs human annotators to confirm or revise. This part could cost quite some label.
>
>   2. Another huge difference between Wikipedia and News is that a single Wiki passage archives the evolution history of a single fact, while the single news passage only contains a fact at a specific time frame. Therefore, a time-sensitive question might need information retrieval to actually find answers from the news corpus. That would bring significant change to the model design, but it’s definitely an interesting new problem to explore.
>
> - Motivation to use Wikipedia:
>
>   The reason to use Wikipedia is to exploit its companion WikiData, where we could easily identify time-evolving facts of our interests. The limitation of our dataset is actually coming from WikiData rather than Wikipedia. Wikipedia is a very huge storage of world facts, it contains lots of interesting time-evolving facts. However, only a small portion of them have their time scope annotated in its companion WikiData. If we could find more effective ways to mine time-evolving facts from Wikipedia, it can scale the dataset up to a broader scope

---

> > ### Comment · Reviewer_bfhP · 2021-10-04
> > **Brief summary after rebuttal**
> >
> > Thank you for the answers and updates in the paper. I'll keep my high rating of this article and believe this is an exciting and well-executed paper.

---

### Official Review · Reviewer_81Tq · 2021-09-20
**Time sensitive QA dataset based on facts from WikiData**

**Rating:** 6
**Confidence:** 4
**Correctness:** seems ok.
**Clarity:** Overall paper is well written.

**Strengths:**

- A new  dataset for training and evaluation of time sensitive QA systems.
- Baseline results using SOTA models for QA over long text.

**Weaknesses:**

- Manual templates for generating questions after fact extraction and verification
- Relation distribution is skewed around 2-3 relation, so a simple template based QA baseline might do great as well.
- The dataset is very similar to TORQUE. Long text can be transformed into short text by extracting relevant context from wiki page.

**Additional Feedback:**

- Probably a comparison with template based QA system is needed.
-How does BigBird and FiD perform on TORQUE/ other timeQA dataset?

**Documentation:**

Yes, data collection and annotation details are clearly mentioned.

**Ethics:**

No.

**Relation To Prior Work:**

Yes discussed.

**Summary And Contributions:**

Authors constructs a new time sensitive QA dataset. They use facts from WikiData for construct this dataset. They take help of crowd workers to filter out noisy facts. Authors claim that this dataset pose QA challenges in terms of both temporal understanding and temporal reasoning. The accuracy using long document QA system FiD for this dataset is far behind human performance.

---

> ### Author Response · Authors · 2021-09-27
> **Author Response**
>
> - Manual templates for generating questions after fact extraction and verification
>
>   1. Using template questions could help us get rid of the complexity of semantic understanding and only focus on the challenge of temporal reasoning. If a model fails, we can safely attribute the error to the lack of temporal reasoning. Even with such simple template questions, the existing SOTA models are not able to do well, which reflects the difficulty in handling time-scoped facts.
>   2. We also agree to the point that some artificial questions are not natural enough, Thus, we release new annotation jobs recently to harvest human-rephrased time-sensitive questions. In total, we obtain 1.2K training and 1K testing human-paraphrased questions (these questions are already in the Github: https://github.com/wenhuchen/Time-Sensitive-QA/tree/main/dataset).
>
> - Relation distribution is skewed around 2-3 relations, so a simple template-based QA baseline might do great as well.
>
>   1. During the dataset construction, we have already downsampled the most frequent relations to balance the distribution.
>   2. In figure 6, we also report fine-grained per-relation accuracy. We found that these overwhelming relations don’t have higher accuracy than the other less popular relations. This means that the bottleneck of our model does not lie in sample complexity. The real bottleneck is the temporal reasoning capability.
>   3. In our newly crowd-sourced questions, we further balanced the distribution across different relations. Please refer to https://github.com/wenhuchen/Time-Sensitive-QA/blob/main/dataset/human_annotated_test.json.
>
> - The dataset is very similar to TORQUE. Long text can be transformed into short text by extracting relevant context from the wiki page.
>   The major difference is that TimeQA contains factoid questions asking about facts in the world, like `Who is the president of US in XXX?`, while TORQUE contains mostly probing questions asking about events, like `What events have already finished?`.
>   1. TORQUE: about events, TimeQA: about evolving world facts.
>   2. TORQUE: contextualized questions, TimeQA: decontextualized, can be used for open-domain question answering.
>   3. TORQUE: coming from TempEval3 domain, TimeQA: from Wikipedia domain.
>   4. TORQUE: short paragraph, TimeQA: over a structured long document
>
> - Template-based QA system:
>
>   I’m not entirely sure what’s a template-based QA system, could you elaborate more on how to build such a template-based QA system? Since the passage is from Wikipedia, it’s not possible to apply any rule on top of this free-form text.

---

### Author Response · Authors · 2021-09-28
**Paper Update (We add human-paraphrased training/test questions to increase diversity; We conduct Open-Domain QA experiments)**

First of all, we thank all the reviewers for their valuable feedback. Based on these comments, we have made significant changes to our current submission, which are listed as follows:

1. We sampled 2000 questions (balanced by relations) from train/test and feed them to Amazon Mechanical Turkers to paraphrase these templates to increase the diversity. The outcome questions are more natural and realistic, which serves as an additional split to help improve our benchmark. We conducted experiments on these human-paraphrased questions and found that **by simply fine-tuning on our template questions, the model can already generalize very well on these paraphrased questions, suffering from only 5% drop**, with further adaptation on the human-paraphrased training questions, the performance gap is further reduced.
2. **We conducted an open-domain QA setting on our human-paraphrased questions**, where the model needs to retrieve the passage from the whole Wikipedia to answer these time-sensitive questions. We follow the BM25+Reader pipeline and found that the retriever score is rather low with HITS@1 roughly 30%. The end QA-task EM accuracy is consequently lower than 15%.
3. We added a more detailed explanation about our annotation guideline in Sec 2.1, and refine the definition of `easy and hard' in Sec 2.2.
4. The newly annotated examples are already released in the GitHub repo: https://github.com/wenhuchen/Time-Sensitive-QA/tree/main/dataset

PDF update:
  1. We add a sub-section to discuss how we collected  human paraphrasing in Sec 2.2
  2. We add new experiment table 3 to show experimental results on human-paraphrased split under both close-domain and open-domain settings.

---

### Decision · Program_Chairs · 2021-10-09

**Decision:**

Accept

**Comment:**

This paper presents a very interesting dataset for time-sensitive reasoning in question answering. All of the reviews are positive, and my reading was also positive. The community, especially those who are interested in NLP and QA, would be quite fascinated to hear about this dataset and get their hands on it.